# Temporal and geographic patterns of stab injuries in young people: a retrospective cohort study from a UK major trauma centre

Paul Vulliamy,[1] Mark Faulkner,[2] Graham Kirkwood,[3] Anita West,[4] Breda O'Neill,[4] Martin P Griffiths,[4] Fionna Moore,[5] Karim Brohi[1,4]

[1]Centre for Trauma Sciences, Queen Mary University of London, London, UK
[2]London Ambulance Service NHS Trust, London, UK
[3]Institute of Health and Society, Newcastle University, Newcastle-upon-Tyne, UK
[4]Trauma Service, The Royal London Hospital, Barts Health NHS Trust, London, London, UK
[5]South East Coast Ambulance Service NHS Foundation Trust, Crawley, UK

**Correspondence to**
Dr Paul Vulliamy;
p.vulliamy@qmul.ac.uk

## ABSTRACT

**Objectives** To describe the epidemiology of assaults resulting in stab injuries among young people. We hypothesised that there are specific patterns and risk factors for injury in different age groups.

**Design** Eleven-year retrospective cohort study.

**Setting** Urban major trauma centre in the UK.

**Participants** 1824 patients under the age of 25 years presenting to hospital after a stab injury resulting from assault.

**Outcomes** Incident timings and locations were obtained from ambulance service records and triangulated with prospectively collected demographic and injury characteristics recorded in our hospital trauma registry. We used geospatial mapping of individual incidents to investigate the relationships between demographic characteristics and incident timing and location.

**Results** The majority of stabbings occurred in males from deprived communities, with a sharp increase in incidence between the ages of 14 and 18 years. With increasing age, injuries occurred progressively later in the day ($r^2$=0.66, p<0.01) and were less frequent within 5 km of home ($r^2$=0.59, p<0.01). Among children (age <16), a significant peak in injuries occurred between 16:00 and 18:00 hours, accounting for 22% (38/172) of injuries in this group compared with 11% (182/1652) of injuries in young adults. In children, stabbings occurred earlier on school days (hours from 08:00: 11.1 vs non-school day 13.7, p<0.01) and a greater proportion were within 5 km of home (90% vs non-school day 74%, p=0.02). Mapping individual incidents demonstrated that the spike in frequency in the late afternoon and early evening was attributable to incidents occurring on school days and close to home.

**Conclusions** Age, gender and deprivation status are potent influences on the risk of violent injury in young people. Stab injuries occur in characteristic temporal and geographical patterns according to age group, with the immediate after-school period associated with a spike in incident frequency in children. This represents an opportunity for targeted prevention strategies in this population.

## INTRODUCTION

Interpersonal violence involving knives is a major public health problem.[1] Reports of

### Strengths and limitations of this study

► Large long-term study focusing on an important public health issue in age groups at the highest risk of knife violence.
► Unlike most previous studies on this topic, this study combines incident-level data on location and timing of assaults with demographic data and clinical outcomes which allows a detailed analysis of the epidemiology of knife violence in specific age groups.
► This study does not provide an insight into the patterns of stab injuries over time because of changes to the trauma system which occurred during the study period.
► The generalisability of these findings to other settings and other forms of interpersonal violence may be limited.

high-profile incidents are rarely absent from mainstream media, although these represent a fraction of the overall incidence. In 2017, 36 998 offences involving knives or other sharp implements were reported in England and Wales, a rise of 26% compared with the previous year.[2] Beyond the immediate physical consequences of knife violence, the psychological and social impact on individuals and communities is substantial.[3 4] Despite intensive efforts at prevention, the incidence of knife crime has increased in recent years.[2 5 6] A detailed understanding of the risk factors for stab injury in specific groups is essential to inform ongoing preventative initiatives.

Young people are the most frequent victims of knife violence.[7] Male teenagers from deprived communities in urban areas are at particularly high risk, with a peak in incidence between the ages of 16 and 24 years.[1 2 8] Stab injuries in children are less common than in adolescents, but also predominantly affect those living in areas with the highest level of socioeconomic deprivation.[9] The extent

to which the timing, location and outcomes of stab injuries vary with age is unknown. A detailed breakdown of the epidemiology of stab injuries in different age groups would identify opportunities for targeted prevention measures in individual populations.

The objective of this study was to characterise the epidemiology of stab injuries among different age groups. We hypothesised that stabbings in children occur in specific temporal and geographical patterns which are distinct from those involving adolescents and young adults.

## METHODS

### Study design and setting

We performed a retrospective cohort study of patients presenting to an urban major trauma centre in London, UK. Our hospital receives approximately 3000 patients requiring trauma team activations per year, of which 25% are now penetrating injuries. The trauma service covers a population of around 3.5 million people and encompasses some of the most deprived regions in the country. All patients who met criteria for trauma team activation between 2004 and 2014 were screened for inclusion. We included patients under the age of 25 who presented to the emergency department after an injury involving a knife or other sharp implement. Accidental injuries and those resulting from deliberate self-harm were excluded. Demographic data, injury characteristics and outcomes were recorded prospectively by a dedicated trauma nurse practitioner. Incident time and location were obtained retrospectively from the regional ambulance service database. Death preventability was determined by local peer review.

### Definitions

To investigate age-group specific characteristics, we subdivided the cohort according to the WHO definitions of childhood (<16 years), late adolescence (16–19 years) and young adulthood (20–24 years).[10] Deprivation status was determined using the Index of Multiple Deprivation score based on home postcode and classified into quintiles according to nationally defined cut-offs. Patients with missing data for home address were excluded from these analyses.

### Data analysis

Data were analysed using Microsoft Excel V.15.3 (Microsoft, California, USA) and Prism V.6.0 (GraphPad, California, USA). Maps were generated using Tableau V.10.1 (Tableau software, Washington, USA). Distances between postcodes were calculated using open access online software (www.freemaptools.com). Continuous data are reported as median with IQR and have been compared with Mann-Whitney U tests or Kruskal-Wallis tests with post-hoc corrections for multiple comparisons. Categorical data are displayed as number and percentage and have been compared with Fisher's exact test. A two-tailed p value less than 0.05 was considered significant throughout.

### Patient and public involvement

No patients were involved in the research design, and no patients were directly involved in the study. Since the conception of this study, we have established a network of patients and public representatives who help guide our ongoing violence-reduction initiatives and will be involved in dissemination of these findings to communities and other institutions.

## RESULTS

Between 2004 and 2014, 3274 victims of assault resulting in penetrating trauma presented to the emergency department at our institution, of whom 1824 (56%) were aged under 25 and were included in the analysis. Of these, 172 (9.4%) were children, 861 (47.2%) were aged 16–19 and 791 (43.4%) were aged 20–24 (table 1). The locations of individual incidents over time is shown in online supplementary video 1. The annual number of presentations increased by an average of 25% each year. A substantial majority of patients (1127/1594, 71%) were from the most deprived quintile while only 1% (15/1594) were from the least deprived quintile, excluding 230 patients who had no recorded home postcode. No major demographic differences were identified across the three age groups, but there was a trend towards higher in-hospital mortality in the paediatric group (7/172, 4.1%) compared with the adolescent and young adult group (26/1652, 1.9%, p=0.08) despite comparable injury severity scores. Preventable or potentially preventable deaths at peer review were significantly more frequent in paediatric patients compared with older adolescents and young adults (5/7 deaths in the paediatric group vs 6/38 in those aged ≥16, p=0.001).

Distinct patterns of injury were observed across the age range within the study cohort. The frequency of stab injuries rose sharply in the late teenage years, reaching a peak at age 18 before gradually declining (figure 1A). The severity of physical injury did not vary with age, and the majority of injuries were classified as non-critical (table 1). There were significant differences in the timing and location of injuries, with younger patients tending to be stabbed earlier in the day and closer to home (figure 1B,C).

To investigate these variations further, we compared patterns of injury in children (<16) with older adolescents and young adults (16–24). Among children, a significant peak in frequency occurred between 16:00 and 18:00 hours, accounting for 22% of all injuries in this age group as compared with 11% in young adults (p<0.01, figure 2A). Young adults were significantly more likely to be stabbed after midnight (16–24 years old 31% vs <16 years old 16%, p<0.01). We next examined the distance from home address to incident in these two age groups and found distinct distributions (figure 2B).

**Table 1** Demographics of the study cohort

| Number of patients | All patients 1824 | <16 years 172 | 16–19 years 861 | 20–24 years 791 | P value |
|---|---|---|---|---|---|
| **Patient characteristics** | | | | | |
| Age, years | 19 (17–21) | 15 (14–15) | 17 (17–18) | 22 (20–23) | <0.001 |
| Male, n (%) | 1772 (97) | 169 (98) | 847 (98) | 756 (96) | 0.002 |
| Index of Multiple Deprivation* | 42 (32–50) | 44 (33–50) | 43 (33–50) | 41 (31–50) | 0.38 |
| Most deprived quintile, n (%)* | 1127/1594 (71) | 116/157 (74) | 564/776 (73) | 447/661 (68) | 0.06 |
| **Injury characteristics** | | | | | |
| Injury Severity Score | 2 (1–9) | 1 (1–9) | 2 (1–9) | 2 (1–9) | 0.39 |
| Severe injury, n (%)† | 225 (12) | 24 (14) | 99 (12) | 102 (13) | 0.51 |
| Multiple injuries, n (%) | 947 (52) | 81 (47) | 456 (53) | 410 (52) | 0.37 |
| Hospital admission, n (%) | 1116 (61) | 103 (62) | 548 (64) | 465 (59) | 0.13 |
| Length of stay, days‡ | 2 (1–4) | 2 (1–4) | 2 (1–4) | 2 (1–4) | 0.61 |
| In-hospital mortality, n (%) | 39 (2) | 7 (4) | 12 (1) | 20 (2) | 0.05 |

Values are median with IQR unless stated.

P values compare age groups with Kruskal-Wallis test for continuous data and $\chi^2$ test for categorical data.

*Excluding 230 patients with missing home postcodes.

†Injury Severity Score >15.

‡Hospital admissions only.

A large proportion of incidents occurred within 1 km of home in both children (35%) and young adults (41%, p=0.08). Children were significantly more likely to be stabbed between 1 and 5 km from home (48% vs 35%, p=0.002) but less likely to be stabbed more than 5 km from home (16% vs 25%, p=0.04).

We hypothesised that the specific pattern of injuries in children was related to school attendance. We therefore subdivided incidents into those occurring on school days and those occurring during school holidays or at weekends. Children were more likely to be stabbed on a school day than the older age group (58% vs 50%, p=0.06). In the paediatric group, stabbings occurred earlier on a school day (hours from 08:00: school day 11.1 (8.6 to 13.9) vs non-school day 13.7 (9.7 to 15.9], p<0.01; figure 2C) and a greater proportion were within 5 km of home (school day 90% vs non-school day 74%, p=0.02; figure 2D). Mapping individual incidents demonstrated that in children the spike in frequency in the late afternoon and early evening was attributable to incidents occurring on school days (figure 3). The majority of stabbings in this time frame on school days occurred within 5 km of home which encompasses the average distance from home to school in children living in London.[11] On non-school days, incidents in children were similar to those in young adults in terms of the timing (hours from 08:00: 13.7 (9.7 to 15.9) vs 13.9 (10.1 to 17.5), p=0.22) and location relative to home (proportion >5 km from home 25.8% vs 26.4%, p=1.0). Stabbings on school days within 5 km of home accounted for 47% of the total injury burden among children, compared with 33% in young adults (p<0.001).

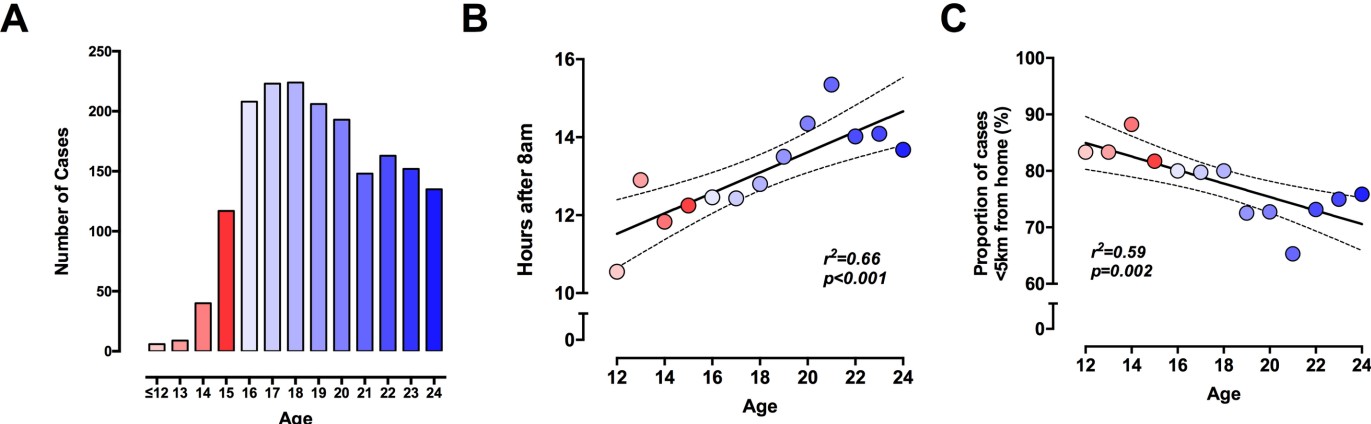

**Figure 1** Age-related variations in the pattern of stab injury. (A) Number of patients by age. (B) Time of injury. (C) Proportion of incidents occurring within 5 km of home. Linear regression line and 95% CIs shown with solid and dashed lines, respectively.

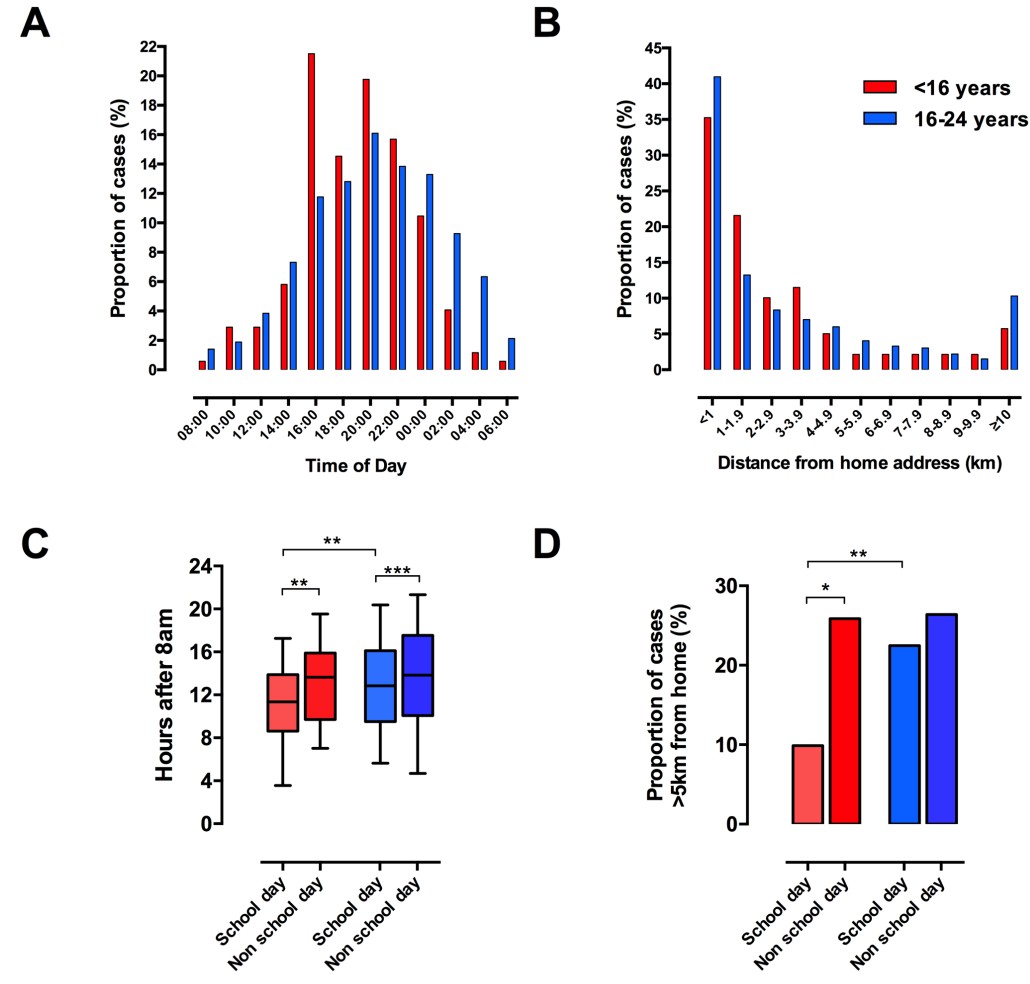

**Figure 2** Injuries in children occur in characteristic patterns and are related to school attendance. (A) Timing of injuries in children (red) and young adults (blue). (B) Distance from home address to incident in children and young adults. (C) Timing of injuries on school days compared with weekends or school holidays. (D) Location of injuries on school days compared with weekends or school holidays. *P<0.05, **P<0.01, ***P<0.001, Kruskal-Wallis test with Dunn's multiple comparisons test.

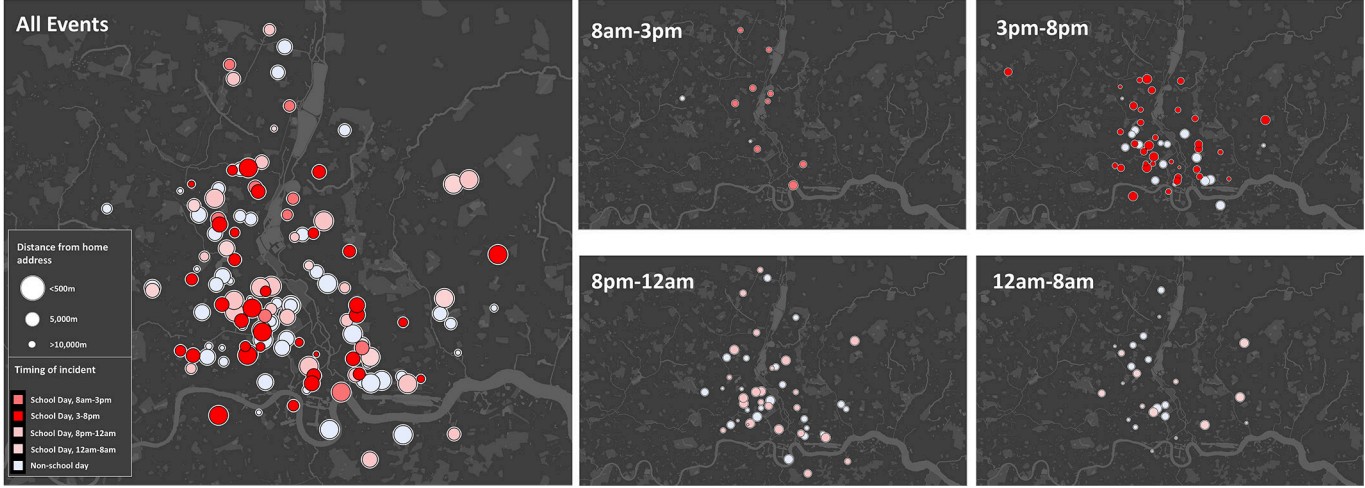

**Figure 3** Incident locations, timing and distance from home in children on school days and non-school days. Circle size is inversely proportional to the distance from incident to home address. Colours indicate incident timing.

## DISCUSSION

In this long-term retrospective cohort study, we have shown that assaults resulting in penetrating injuries occur in distinct age-related patterns. Specifically, the period immediately after school accounts for a large proportion of incidents in children, and these predominantly occur close to home and school. This represents an opportunity for targeted preventative strategies in this population.

At both an individual and community level, knife crime has major physical, psychological and social consequences.[1 12] The incidence of interpersonal violence involving knives has progressively increased in the UK in recent years, reaching a 7-year high in 2017, and anecdotal reports suggest that assaults resulting in multiple injuries from multiple assailants are also on the rise.[2 7] However, multimodal preventative strategies can produce dramatic reductions in weapons offences and injuries among young people.[13] This is exemplified by the success of violence-reduction initiatives in Glasgow which have resulted in consistent and substantial decreases in knife crime in a country labelled the most violent in the developed world by the United Nations in 2005.[14 15] Aggressive law enforcement formed the initial basis of this programme, including legislation to impose mandatory sentences for knife possession, increased duration of sentences and widespread use of the 'stop and search' strategy. In isolation, such strategies have had limited effect in other settings and may increase tensions between communities and law enforcement.[16 17] Crucially, in the Glasgow example, this approach is coupled with a range of educational and behavioural programmes which raise awareness of the consequences of knife violence.[13] Addressing the broader socioeconomic factors associated with violence is another cornerstone of many successful prevention initiatives. Neighbourhood deprivation and concentrated poverty have consistently been shown to be independent risk factors for involvement in violent crime,[18] an association which is clearly reflected in our cohort. There is ample evidence that community-based interventions to reduce environmental contributors to violence and minimise inequality can reduce the incidence of offending, violent injury and incarceration among young people.[19–21] It is clear that a multifaceted approach with sustained investment from government and the community is required for effective violence reduction.[22 23]

Our results provide detailed age-specific information regarding the timing and location of stab injuries which have a number of potential implications for targeted violence-reduction strategies. The sharp increase in stab injuries between the ages of 14 and 16 suggests that educational programmes and other preventative interventions are best delivered in primary or early secondary education. Given the peak in incidents at the end of the school day, an attractive option is staggered release times from school which could be coupled with a visible deterrent by law enforcement at transport hubs, eateries and other areas of pupil congregation after school.

Combating weapons carriage through a 'stop and search' strategy, which remains a hotly debated issue, will be better informed by accurate incident data such as that presented here. However, these direct approaches will only produce a sustained reduction in knife crime when delivered in the context of a co-ordinated plan to combat violent behaviour and its root causes.

From a clinical perspective, the majority of stab injuries in our cohort resulted in relatively minor physical injuries, and deaths were infrequent. However, over half of all stabbings resulted in multiple injuries. This is more than double the frequency observed in a study conducted in our catchment area 30 years ago[24] which supports anecdotal observations of increasing intensity of violence involving knives. We found that children had a higher overall mortality and a higher frequency of potentially preventable deaths compared with young adults despite comparable injury severity scores. Further study is required to determine the reasons for this observation and whether a similar trend is evident in other trauma systems and in other injury patterns. Although the implementation of regional trauma networks in England and Wales has produced substantial improvements in outcomes from major injury in adults, similar evidence in children is currently lacking and is the subject of an ongoing multicentre investigation.[25]

The limitations to our study include its observational design, most notably in that the apparent relationship between paediatric stab injuries and school attendance represents an association only. The generalisability of our findings to other geographical areas and other forms of interpersonal violence may be limited. Because we only included patients meeting criteria for trauma team activation, our study may have missed those attending with more minor injuries. Owing to the healthcare-based setting of this study, we were not able to consider the demographics of the assailants and the victims in this analysis. In addition, our registry did not allow detailed analysis of behavioural patterns, violent recidivism and gang involvement in individual patients. These gaps in the present study are currently being investigated prospectively within our trauma system. Finally, because of changes in our regional trauma system during the study period, we were not able to comment on trends in incidence over time.

Our study illustrates and reiterates the potent influence of deprivation, age and gender on the risk of violent injury. We have demonstrated that there are age-specific epidemiological patterns of stabbings among young people and identify specific targets for focused violence-reduction strategies in children. Long-term, multiagency interventions are essential to drive sustained reductions in interpersonal violence and will be better informed by the recognition of knife crime as a pressing public health issue.

**Contributors** PV, FM and KB conceived the study. PV, MF, AW, GK, BON, MPG and KB contributed to data collection and analysis. PV and KB wrote the initial

draft of the manuscript. All authors contributed to critical revisions of subsequent manuscript drafts and approve of the final version.

**Funding** The authors have not declared a specific grant for this research from any funding agency in the public, commercial or not-for-profit sectors.

**Competing interests** None declared.

**Patient consent** Not required.

**Ethics approval** NRES Committee South Central - Berkshire B (reference 15/SC/0547).

**Provenance and peer review** Not commissioned; externally peer reviewed.

**Data sharing statement** No additional data available.

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
