## [Reviewer comments · BMJ Open]

ARTICLE DETAILS

TITLE (PROVISIONAL)	Temporal and geographic patterns of stab injuries in young people: a retrospective cohort study from a UK major trauma centre
AUTHORS	Vulliamy, Paul; Faulkner, Mark; Kirkwood, Graham; West, Anita; O'Neill, Breda; Griffiths, Martin; Moore, Fionna; Brohi, Karim

VERSION 1 – REVIEW

REVIEWER	Justin Myers University of North Carolina at Chapel Hill, USA
REVIEW RETURNED	15-May-2018

GENERAL COMMENTS	1. Can you include more about this trauma center, such as visits per year, possible catchment area, etc? This will help readers with generalizability and would better define the burden in this hospital system.2. What percent of these incidents were trauma activations? Also, do you need to report these non-significant p values in Table 1, or could you just state in the text that injury characteristics among this cohort were not statistically significant. I would defer this to BMJ's editor preference.3. I understand you are dealing with a limited dataset (the trauma registry). However, further information would be important in using this information for developing a violence reduction strategy, such as: ED recidivism of these patients, social factors such as % employed, % graduated from secondary school, % in university, % incarcerated, % involved in gang, % involved in after/paraschool activities (sports, church/faith-based, etc). And finally, understanding WHY or what is prompting this violence: social media, gang activity, lifestyle or racial profiling, etc, etc. Finally, who is the assailant: peer groups or older ages? Can you share anything further about injury patterns from the trauma registry?4. Does this limit the ability to show volume of stabbings per year over time? It would be helpful to at least report the volume of stabbings that have occurred over the past year - to understand the current burden of this violence. (ie, were half of these stabbings in the past year? That would certainly increase the urgency of an intervention)5. Perhaps outside the scope of this paper, but there is a ton of work being done in the field of youth violence prevention/reduction. I would add a little more about what a successful violence reduction implementation would look like...and the investment required by the entire community and government. While not an advocacy paper per se, it is a platform to highlight for lawmakers and the community's responsibility to reduce this violence. Not unlike Glasgow, cities in the US are taking a multifaceted approach that requires many tiers of community and government engagement: https://youth.gov/youth-topics/preventing-youth-violence/spotlight-forum-communities. I
---

	agree that it is much more than just implementing a “stop and search” strategy. This article talks about a multi-pronged approach. Miao T, Umemoto K, Gonda D, & Hishinuma ES (2011). Essential elements for community engagement in evidence-based youth violence prevention. American Journal of Community Psychology, 48: 120-132. 6. This graph of ISS is not really helpful — all the error bars appear to overlap and I don't think it adds much (a restatement of Table 1). 7. The usage of GIS to understand the time/geospatial burden of this pediatric violence if effective. However, this graph is hard to read. As long as this is published with sufficient size to read- it would work.
--	--

REVIEWER	Angela Sauaia, MD, PhD University of Colorado Denver, USA
REVIEW RETURNED	01-Jul-2018

GENERAL COMMENTS	Outside the Americas, where firearm-related violence remains a major public health problem, most homicides in young persons are due to injuries by sharp objects . Accordingly, Vulliamy and colleagues, from the UK, focused on violent stab injuries in this epidemiological assessment of young patients (<25 years of age) presenting to an urban trauma center. Although not a population-based study, it represents the group of patients who accessed a major trauma care system, thus a high-risk population. While overall case-fatality rate was relatively low (<5%), the burden of these injuries was substantial, with close to two-thirds of the patients requiring hospitalization. The authors confirmed previous findings showing that deprivation is a major risk factor for violent stab injuries. Indeed, poverty remains the major risk factor of most diseases and injuries. Of note, close to 10% of the victims were children less than 16 years of age, among whom the mortality was slightly (albeit not significantly) higher than among individuals 16-25 years old. Close to half of the events involved multiple injuries, characterizing extreme violence. The major contribution of this study is the finding that among young children, most incidents occurred right after school and close to home, information that can be used in targeted interventions. As noted by the authors, the absence of reliable, accessible data sources on the perpetrators and circumstances surrounding the incidents limits the translation of violence-relayed research into effective interventions. Concerted, focused efforts should be placed on adding these crucial pieces of information to inform future prevention programs. World Health Organisation. European report on preventing violence and knife crime among young people. 2017. http://www.euro.who.int/__data/assets/pdf_file/0012/121314/E94277.pdf; United Nations Office on Drugs and Crime: 2011 Global study on homicide at http://www.unodc.org/documents/data-and-analysis/statistics/Homicide/Globa_study_on_homicide_2011_web.pdf. Both accessed June 30th , 2018.
---

VERSION 1 – AUTHOR RESPONSE

Reviewer #1

Dear Authors,

This is a very interesting paper with a nice GIS and statistical assessment of this community burden of violence. It unearths more questions that are ripe for further study to understand the problem in this area. I have uploaded the pdf with comments and the comments are restated below.

I appreciate the opportunity to review it.

Sincerely,
Justin Myers

1. Can you include more about this trauma center, such as visits per year, possible catchment area, etc? This will help readers with generalizability and would better define the burden in this hospital system.

Thank you. We have included a more detailed description of our centre and its catchment in the first paragraph of the methods section (paragraph 1), and retitled the initial subsection of the methods to 'study design and setting'.

2. What percent of these incidents were trauma activations? Also, do you need to report these non-significant p values in Table 1, or could you just state in the text that injury characteristics among this cohort were not statistically significant. I would defer this to BMJ's editor preference.

All of the patients in the study were trauma team activations. This is stated in the methods section (paragraph 1). We have included a statement in the limitations (discussion, paragraph 5) to emphasise that patients with minor injuries have not been included.

Regarding the reporting of non-significant p-values in table 1, it appears that this is acceptable practice for BMJ Open but we are of course happy to remove at the editor's discretion.

3. I understand you are dealing with a limited dataset (the trauma registry). However, further information would be important in using this information for developing a violence reduction strategy, such as: ED recidivism of these patients, social factors such as % employed, % graduated from secondary school, % in university, % incarcerated, % involved in gang, % involved in after/paraschool activities (sports, church/faith-based, etc). And finally, understanding WHY or what is prompting this violence: social media, gang activity, lifestyle or racial profiling, etc, etc. Finally, who is the assailant: peer groups or older ages? Can you share anything further about injury patterns from the trauma registry?

Thank you for these important comments. Unfortunately this data is not collected as part of our registry. We have added to the limitations section (discussion, paragraph 5) to emphasise this. We agree that the characteristics of the assailant are extremely important but we feel this data is beyond the scope of the current study. We are currently performing a prospective study to profile both patients and assailants in detail which will encompass many of the characteristics you describe and inform violence reduction strategies.

4. Does this limit the ability to show volume of stabbings per year over time? It would be helpful to at least report the volume of stabbings that have occurred over the past year - to understand the current burden of this violence. (ie, were half of these stabbings in the past year? That would certainly increase the urgency of an intervention)

We have edited the results section to describe the trend in stabbings over time (results, paragraph 1). We have also altered the discussion to describe to the current burden of violence and the most recent trends in our catchment area (discussion, paragraph 2). Because of changes in our trauma system during the study period we have not focused on annual trends in detail in this manuscript but this is the subject of ongoing work by ourselves and other agencies.

5. Perhaps outside the scope of this paper, but there is a ton of work being done in the field of youth violence prevention/reduction. I would add a little more about what a successful violence reduction implementation would look like...and the investment required by the entire community and government. While not an advocacy paper per se, it is a platform to highlight for lawmakers and the community's responsibility to reduce this violence. Not unlike Glasgow, cities in the US are taking a multifaceted approach that requires many tiers of community and government engagement: <https://youth.gov/youth-topics/preventing-youth-violence/spotlight-forum-communities>. I agree that it is much more than just implementing a "stop and search" strategy. This article talks about a multi-pronged approach. Miao T, Umemoto K, Gonda D, & Hishinuma ES (2011). Essential elements for community engagement in evidence-based youth violence prevention. *American Journal of Community Psychology*, 48: 120-132.

Thank you for this comment – we could not agree more and we have attempted to emphasise the need for a sustainable, multi-agency approach in the discussion and conclusion. We have added to these sections (discussion, paragraphs 2,5,6) and included additional references (#22 and #23) in an attempt to further emphasise these points.

6. This graph of ISS is not really helpful — all the error bars appear to overlap and I don't think it adds much (a restatement of Table 1).

We agree and have removed this panel from figure 1.

7. The usage of GIS to understand the time/geospatial burden of this pediatric violence if effective. However, this graph is hard to read. As long as this is published with sufficient size to read- it would work.

Thank you. We have included an enlarged version in our resubmission and are happy to liaise with the editorial and production team to ensure that a suitable version of the figure is available.

Reviewer #2

Outside the Americas, where firearm-related violence remains a major public health problem, most homicides in young persons are due to injuries by sharp objects. Accordingly, Vulliamy and colleagues, from the UK, focused on violent stab injuries in this epidemiological assessment of young patients (<25 years of age) presenting to an urban trauma center. Although not a population-based study, it represents the group of patients who accessed a major trauma care system, thus a high-risk population. While overall case-fatality rate was relatively low (<5%), the burden of these injuries was substantial, with close to two-thirds of the patients requiring hospitalization. The authors confirmed previous findings showing that deprivation is a major risk factor for violent stab injuries. Indeed, poverty remains the major risk factor of most diseases and injuries. Of note, close to 10% of the victims were children less than 16 years of age, among whom the mortality was slightly (albeit not significantly) higher than among individuals 16-25 years old. Close to half of the events involved multiple injuries, characterizing extreme violence. The major contribution of this study is the finding that among young children, most incidents occurred right after school and close to home, information that can be used in targeted interventions. As noted by the authors, the absence of reliable, accessible data sources on the perpetrators and circumstances surrounding the incidents limits the translation of violence related research into effective interventions. Concerted, focused efforts should be placed on adding these crucial pieces of information to inform future prevention programs.